# Statistical Analysis of Nearest Neighbor Methods for Anomaly Detection

**Xiaoyi Gu**[1], **Leman Akoglu**[2], **Alessandro Rinaldo**[1]
[1]Department of Statistics and Data Science, Carnegie Mellon University
[2]Heinz College of Information Systems and Public Policy, Carnegie Mellon University
{xgu1,lakoglu}@andrew.cmu.edu, arinaldo@cmu.edu

## Abstract

Nearest-neighbor (NN) procedures are well studied and widely used in both supervised and unsupervised learning problems. In this paper we are concerned with investigating the performance of NN-based methods for anomaly detection. We first show through extensive simulations that NN methods compare favorably to some of the other state-of-the-art algorithms for anomaly detection based on a set of benchmark synthetic datasets. We further consider the performance of NN methods on real datasets, and relate it to the dimensionality of the problem. Next, we analyze the theoretical properties of NN-methods for anomaly detection by studying a more general quantity called distance-to-measure (DTM), originally developed in the literature on robust geometric and topological inference. We provide finite-sample uniform guarantees for the empirical DTM and use them to derive misclassification rates for anomalous observations under various settings. In our analysis we rely on Huber's contamination model and formulate mild geometric regularity assumptions on the underlying distribution of the data.

## 1   Introduction

Anomaly detection is the process of detecting instances that deviate significantly from the other sample members. The problem of detecting anomalies can arise in many different applications, such as fraud detection in financial transactions, intrusion detection for security systems, and various medical examinations.

Depending on the availability of data labels, there are multiple setups for anomaly detection. The first is the supervised setup, where labels are available for both normal and anomalous instances during the training stage. Because of its similarity to the standard classification setup, numerous classification methods with good empirical performance and well-studied theoretical properties can be adopted. The second setup is the semi-supervised setup, where training data only comprise normal instances and no anomalies. This setup is widely used in the intrusion detection literature. Well-known methods with theoretical guarantees include $k$NNG [1], BP-$k$NNG [2] and BCOPS [3], with the first two methods developed based on the geometric entropy minimization (GEM) principle proposed in [1], and the third on conformal prediction. Methods under this setups are essentially targeting the estimation of high density regions, and treating low density points as anomalies. The third setup is the unsupervised setup, which is the most flexible yet challenging setup. For the rest of the paper, we will only focus on this setup and do not assume any prior knowledge on data labels.

Many empirical methods have been developed in the unsupervised setup, which can be roughly classified into four categories: density based methods such as the Robust KDE (RKDE) [4], Local Outlier Factor (LOF) [5], and mixture models (EGMM); distance based methods such as $k$NN [6] and Angle-based Outlier Detection (ABOD) [7]; model based methods such as the one-class SVM (OCSVM) [8], SVDD [9], and autoencoders [10]; ensemble methods such as Isolation Forest

(IForest) [11] and LODA [12]. In practice, ensemble methods are often favored for their computational efficiency and robustness to tuning parameters, yet there is little theoretical understanding of how and why these algorithms work. Recent work on NN-methods combine $k$NN with sub-sampling [13] [14] or bagging [15] [16], and show that such methods are comparable to the other state-of-the-art methods, both in performance and computational efficiency. Moreover, some theoretical results [14] [17] have been developed on how these methods work.

In this paper, we focus on studying NN-methods in the unsupervised setting, without any sub-sampling or bagging. We begin with an empirical analysis of NN-methods on a set of synthetic benchmark datasets and show that they compare favorably to the other state-of-the-art algorithms. We further discuss their performance on real datasets and relate it to the dimensionality of the problem. Next, we provide statistical analysis of NN-methods by analyzing the distance-to-a-measure (DTM) [18], a generalization to the NN scheme. The quantity was initially raised in the robust topological inference literature, in which DTM proves to be an effective distance-like function for shape reconstruction in the presence of outliers [19]. We give finite sample uniform guarantees on the empirical DTM, and also demonstrate how DTM classifies the anomalies, under suitable assumptions on the underlying distribution of the data. Our theoretical results differ, both in assumptions and goals, from those provided in [17] [14] significantly, and provide complementary insights into the performance of NN-based methods both for anomaly detection and for more general tasks.

## 2 Empirical Performance of NN-methods

Two versions of the NN anomaly detection algorithms have been proposed: $k^{\text{th}}$NN [20] and $k$NN [6]. $k^{\text{th}}$NN assigns anomaly score of an instance by computing the distance to its $k^{\text{th}}$-nearest-neighbor, whereas $k$NN takes the average distance over all $k$-nearest-neighbors. Both methods are shown to have competitive performance in various comparative studies [21, 22, 12, 23]. In particular, the comparative study developed by Goldstein and Uchida [21] is the one of most comprehensive analysis to date that includes the discussion of NN-methods and, at the same time, aligns with the unsupervised anomaly detection setup. However, the authors omit the analysis of ensemble methods, some of which are considered as state-of-the-art algorithms (e.g., IForest and LODA). Emmott et al. [24] constructed a large corpus (over 20,000) of synthetic benchmark datasets that vary across multiple aspects (e.g., clusteredness, separability, difficulty, etc). The authors evaluate the performance of eight top-performing algorithms, including IForest and LODA, but omit the analysis of NN-methods.

In this section, we provide a comprehensive empirical analysis of NN-methods by comparing $k$NN, $k^{\text{th}}$NN, and $\text{DTM}_2$[1] to IForest, LOF and LODA on (1) the corpus of synthetic datasets developed in [24], (2) 23 real datasets from the ODDS library [25], and (3) 6 high dimensional datasets from the UCI library [26]. The code for all our experiments are publicly available[2]. In general, no one methodology should be expected to perform well in all possible scenarios. In Appendix D we present different examples in which IForest, LODA, LOF and $\text{DTM}_2$ perform very differently. For all our experiments, we set the following hyperparameters for our models: sub-sampling size $= 256$ and the number of trees $= 100$ for IForest; $k = 0.03 \times$ (sample size) for all distance based methods for comparable results; for LODA, we use 100 projections with each projection using approximately $\sqrt{d}$ features. The discussion on the robustness of distance-based methods to the choice of hyperparameter $k$ can be found at [27] [28].

### 2.1 Comparison on Benchmark Datasets

First, we complement Emmott et al.'s study [24] by extending it to NN-based detectors. First, we calculate the ROC-AUC (AUC) and Average Precision (AP) scores for each method on each benchmark, and compute their respective quantiles on the empirical distributions for AUC and AP scores (refer to Appendix E in [24] for more details on treating AUC and AP as random variables). We say that an algorithm fails on a benchmark with metric AUC (or AP) at significance level $\alpha$ if the computed AUC (or AP) quantiles are less than $(1 - \alpha)$. Then, the failure rate for each algorithm is found as the percentage of failures over the entire benchmark corpus. The failure rate gives a better

Table 1: Algorithm Failure Rate with Significance Level $\alpha = 0.001$.

|         | AUC    | AP     | Either |
|---------|--------|--------|--------|
| ABOD    | 0.5898 | 0.6784 | 0.7000 |
| IForest | **0.5520** | **0.6514** | **0.6741** |
| LODA    | 0.6187 | 0.6955 | 0.7194 |
| LOF     | 0.6016 | 0.7071 | 0.7331 |
| RKDE    | 0.6122 | 0.7030 | 0.7194 |
| OCSVM   | 0.7218 | 0.7342 | 0.7969 |
| SVDD    | 0.8482 | 0.8868 | 0.9080 |
| EGMM    | 0.6188 | 0.7146 | 0.7303 |
| $k$NN   | 0.5646 | 0.6744 | 0.6960 |
| $k^{\text{th}}$NN | 0.5831 | 0.6886 | 0.7100 |
| $\text{DTM}_2$ | 0.5669 | 0.6761 | 0.6977 |

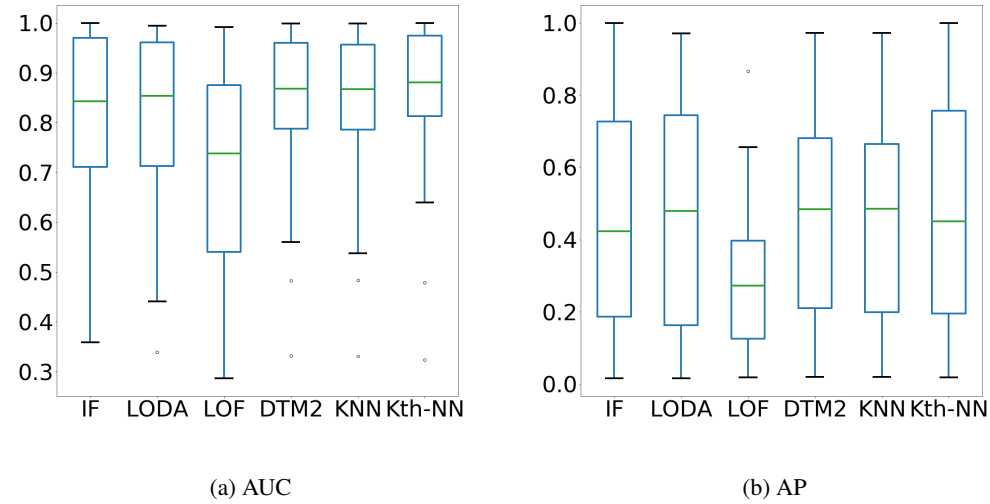

(a) AUC            (b) AP

Figure 1: Boxplots for AUC and AP scores on 23 real datasets.

measure of the overall performance of different methods across the entire benchmark datasets than the average AUC (or AP) scores, as it takes into account of the difficulty of each dataset.

The results are shown in Table 1, where the top section is copied from [24] and the bottom section shows the failure rates we obtained for $k$NN, $k^{\text{th}}$NN, and $\text{DTM}_2$. The "Either" column indicates that the benchmarks fail under at least one of the two metrics. Among all methods, IForest gives the lowest failure rates (boldfaced) for all three metrics. $k$NN and $\text{DTM}_2$ turn out to be next-best top performers, falling marginally behind IForest.

## 2.2 Comparison on Datasets from the ODDS library

Next, we compare the performance of IForest, LODA, LOF, $\text{DTM}_2$, $k$NN and $k^{\text{th}}$NN on 23 real datasets from the ODDS library [25]. Figure 1 presents the overall distributions of AUC and AP scores of the five methods as boxplots. It appears that all methods, except for LOF, have comparable performance, and we further verified this claim via pairwise Wilcoxon signed-rank tests between methods, which showed no statistically significant difference at level 0.05. The full performance table (Table 3) is given in the Appendix, with the last row of the table showing the average rank of each method.

Table 2: AUC and AP performance on high dimensional datasets

| AUC | $n$ | $d$ | IForest | LODA | LOF | $k$NN | $k^{\text{th}}$NN | DTM$_2$ | DTMF$_2$ |
|---|---|---|---|---|---|---|---|---|---|
| gisette | 3850 | 4970 | 0.5023 | 0.5176 | 0.6753 | 0.5696 | 0.5429 | 0.5692 | 0.7051 |
| isolet | 4886 | 617 | 0.5485 | 0.5421 | 0.7330 | 0.6810 | 0.6480 | 0.6796 | 0.7645 |
| letter | 4586 | 617 | 0.5600 | 0.5459 | 0.7846 | 0.7162 | 0.6826 | 0.7149 | 0.8096 |
| madelon | 1430 | 500 | 0.5327 | 0.5427 | 0.5450 | 0.5608 | 0.5552 | 0.5607 | 0.5546 |
| cancer | 385 | 30 | 0.9528 | 0.9626 | 0.8097 | 0.9780 | 0.9756 | 0.9773 | 0.6937 |
| ionosphere | 242 | 33 | 0.9265 | 0.9118 | 0.9450 | 0.9832 | 0.9803 | 0.9824 | 0.9372 |

(a) AUC

| AP | $n$ | $d$ | IForest | LODA | LOF | $k$NN | $k^{\text{th}}$NN | DTM$_2$ | DTMF$_2$ |
|---|---|---|---|---|---|---|---|---|---|
| gisette | 3850 | 4970 | 0.0877 | 0.0907 | 0.1628 | 0.1093 | 0.1015 | 0.1092 | 0.1723 |
| isolet | 4886 | 617 | 0.1005 | 0.1003 | 0.2343 | 0.2074 | 0.1846 | 0.2070 | 0.2458 |
| letter | 4586 | 617 | 0.0956 | 0.0980 | 0.2921 | 0.2328 | 0.2054 | 0.2319 | 0.3010 |
| madelon | 1430 | 500 | 0.1067 | 0.0974 | 0.1171 | 0.1209 | 0.1181 | 0.1209 | 0.1166 |
| cancer | 385 | 30 | 0.6274 | 0.8277 | 0.3121 | 0.8813 | 0.8840 | 0.8864 | 0.2800 |
| ionosphere | 242 | 33 | 0.7222 | 0.7438 | 0.6058 | 0.8903 | 0.8801 | 0.8868 | 0.6105 |

(b) AP

## 2.3 Effect of the dimension

We then take a closer look at the performance of IForest, LODA, LOF, DTM$_2$, $k$NN and $k^{\text{th}}$NN when the data is high dimensional. Additionally, we include the analysis of DTMF$_2$ in our experiments, a quantity defined as the inverse ratio of the DTM$_2$ of a point and the average DTM$_2$ of its $k$-nearest neighbors. DTMF$_2$ can be interpreted as a LOF version of DTM$_2$ and is described in the Appendix A. We consider six high dimensional real datasets from the UCI library [26] (see [12] for details) and compute the AUC and AP scores for each algorithm. The results are presented in Table 2. The $n$ and $d$ columns stand for the number of samples and dimension of the datasets. On datasets *gisette*, *isolet* and *letter*, the performance of IForest and LODA have been significantly downgraded; the NN-methods give somewhat better performance, whereas LOF and DTMF$_2$ are showing significantly stronger performance. However, on datasets *cancer* and *ionoshphere*, where dimensions are slightly lower, the situations are reversed, with LOF and DTMF$_2$ giving significantly worse performance than the others. This is consistent with our findings in Section 2.2. The deficiency of IForest in high dimensions is expected, as the IForest trees are generated by random partitioning along a randomly selected feature. However, in high dimensions, there is a high probability that a large number of features are neglected in the process. From another perspective, [29] discusses the various effects of dimensionality in the context of anomaly detection. In particular, the authors describe a concentration effect of distances in high dimensions, which has a negative effect on IForest, or any other methods that rely on pairwise distances of points for computation of anomaly scores. NN-methods, on the other hand, are somewhat more robust in high dimensions, as the rankings of distance values are still feasible.

Overall, our experiments show that IForest and NN-methods are the top two methods with excellent overall performance on both low dimensional synthetic and real datasets. However, NN-methods exhibit better performance than IForest when the data is high dimensional. In the following sections, we provide a theoretical understanding of how the NN-methods work under the anomaly detection framework.

## 3 Theoretical Analysis

In this section we formalize the settings for a simple yet natural anomaly detection problem based on the classic Huber-contamination model [30, 31], whereby a target distribution generating normal observations is corrupted by a distribution from which anomalous observations are drawn. We introduce the notion of distance-to-a-measure (DTM) [18], as an overall functional of the data based

on nearest neighbors statistics and provide finite sample bounds on the empirical nearest neighbor radii and on the rates of consistency of the DTM in the supremum norm. These theoretical guarantees are novel and may be of independent interest. Finally, we derive conditions under which DTM-based methods provably separate normal and anomalous points, as a function of the level of contamination and the separation between the normal distribution and the anomalous distribution. All the proofs are given in the Appendix B.

## 3.1 Problem Setup

We assume we observe $n$ i.i.d. realizations $\mathbb{X}_n = (X_1, \ldots, X_n)$ from a distribution $P$ on $\mathbb{R}^d$ that follows the Huber contamination model [30, 31]

$$P = (1 - \varepsilon)P_0 + \varepsilon P_1,$$

where $P_0$ and $P_1$ are, respectively, the underlying distribution for the normal and anomalous instances, and $\varepsilon \in [0, 1)$ is the proportion of contamination. Letting $S_0$ and $S_1$ be the support of $P_0$ and $P_1$, respectively, we further assume that $S_0 \cap S_1 = \emptyset$. The distributions $P_0$ and $P_1$, their support and the level of contamination $\varepsilon$ are unknown.

Our goal is to devise a procedure that is able to discriminate the normal observations $X_i$'s belonging to $S_0$, from the anomalous ones, falling in the set $S_1$. Since we will be focusing exclusively on NN methods, we will begin by introducing a population counterpart to the notion of $k$th nearest neighbor. Throughout the article, for any $x \in \mathbb{R}^d$ and $r > 0$, $B(x, r)$ denotes the closed Euclidean ball of radius $r$ centered at $x$.

**Definition 3.1** ($p$-NN radius)**.** *Let $p \in (0, 1)$. For any $x$, define $r_p(x)$ to be the radius of the smallest ball centered at $x$ with $P$-probability mass at least $p$. Formally,*

$$r_p(x) = \inf\{r > 0 : P(B(x, r)) \geq p\}.$$

Naturally, the empirical $p$-NN radius is defined as

$$\hat{r}_p(x) = \inf\{r > 0 : P_n(B(x, r)) \geq p\},$$

where $P_n$ is the empirical measure that puts mass $1/n$ on each $X_i$. Setting $k = \lceil np \rceil$, $\hat{r}_p(x)$ is simply the $k$th-nearest neighbor radius of the point $x$ with respect to the sample $(X_1, \ldots, X_n)$. Thus,

$$P_n(B(x, \hat{r}_p(x)) = \frac{1}{n} \left| \{X_1, \ldots, X_n\} \cap B(x, \hat{r}_p(x)) \right| = \frac{k}{n}.$$

We will impose the following, mild regularity assumptions on the distribution $P$:

- **Assumption (A0):**
  The sets $S_0$ and $S_1$ have diameters bounded by some $L > 0$, and are disjoint from each other.

- **Assumption (A1):**
  There exist positive constants $C = C(P)$ and $\nu_0 = \nu_0(P)$ such that for all $0 < \nu < \nu_0$ and $\gamma \in \mathbb{R}$,
  $$|P(B(x, r_p(x) + \gamma)) - P(B(x, r_p(x)))| \leq \nu \Rightarrow |\gamma| < C\nu,$$
  for $P$-almost every $x$.

- **Assumption (A2):**
  $P_0$ satisfies the **(a,b)-condition**: For $b > 0$, for any $x \in S_0$, there exist $a = a(x) > 0$, and $r > 0$ such that $P_0(B(x, r)) \geq \min\{1, ar^b\}$.

Intuitively, assumption (A1) implies that $P$ has non-zero probability content around the boundary of $B(x, r_p(x))$. Observing further that the function $r \in \mathbb{R}_+ \mapsto F_x(r) = P(B(x, r))$ is the c.d.f. of the random variable $\|X - x\|$, where $X \sim P$, then a sufficient condition for (A1) to hold is that, uniformly over all $x$, $F_x$ has its derivative uniformly bounded away from zero in a fixed neighborhood of $r_p(x)$. This condition, originally formulated in [19] to derive bootstrap-based confidence bands for the DTM function, appears to be a natural regularity assumption in the analysis of NN-type methods. When $a(x) = a$ for all $x \in S_0$, assumption (A2) reduces to a widely used condition in the literature on statistical inference for geometric and topological data analysis [32, 33]. Such condition requires

the support of $P_0$ to not locally resemble a lower dimensional set; in particular, it prevents $S_0$ from having thin ridges or outward cusps. When (A2) is violated, it becomes impossible to estimate $S_0$, no matter the size of the sample. The parameter $b$ can be interpreted as the intrinsic dimension of $P$. In particular, if $P$ admits a strictly positive density on a $D$-dimensional smooth manifold, then it can be shown that $b = D$.

**Definition 3.2** (DTM [18]). *The distance-to-a-measure (DTM) with respect to a probability distribution $P$ with parameter $m \in (0,1)$ and power $q \geq 1$ is defined as*

$$d(x) = d_{P,m,q}(x) = \left( \frac{1}{m} \int_0^m r_p(x)^q \, dp \right)^{1/q}. \tag{1}$$

*When $q = \infty$, we set $d(x) = d_{P,m,\infty}(x) = r_m(x)$.*

It is immediate from the definition that a point $x \in \mathbb{R}^d$ has a small DTM value $d(x)$ if its $p$-NN radii, when averaged across all $p \in (0, m)$ are small. Intuitively, $d(x)$ can be thought of as a measure of the distance of $x$ from the bulk of the mass of the probability distribution $P$ at level of accuracy specified by the parameter $m$. The choice of the parameter $q$ allows to weight differently the impact of large versus small $p$-NN radii.

By substituting $r_p(x)$ with $\hat{r}_p(x)$ in (1), the empirical DTM can be seen to be

$$\hat{d}(x) = d_{P_n,m,q}(x) = \left( \frac{1}{k} \sum_{X_i \in N_k(x)} \|X_i - x\|^q \right)^{1/q},$$

where $k = \lceil mn \rceil$ and $N_k(x)$ denotes the set of $k$-nearest neighbors to $x$ in the sample. Different values of $q \geq 1$ yield different NN-functionals. In particular, the empirical DTM with $q = 1$ is equivalent to the $k$NN method, and the empirical DTM with $q = \infty$ is equivalent to $k^{\text{th}}$NN. The notion of DTM was initially introduced in the geometric inference literature [19], where DTM was developed for shape reconstruction under the presence of outliers. The DTM is known to have several nice properties: it is 1-Lipschitz and it is robust with respect to perturbations of the original distributions with respect to the Wasserstein distance. The case of $q = 2$ is special: the corresponding DTM, denoted below as $\text{DTM}_2$, is also semi-concave and distance-like, and admits strong regularity conditions on its sub-level sets. Chazal et al. [19] have also derived the limiting distribution and a confidence band for the DTM.

## 3.2 Uniform bounds for $\hat{r}_p$ and $\hat{d}$

In this section we derive finite sample bounds on the deviation of $\hat{r}_p$ and $\hat{d}$ from $r_p$ and $d_{P,m,q}$, respectively, that hold uniformly over all $x \in \mathbb{R}^d$ or only over the sample points. These theoretical guarantees are, to the best of our knowledge, novel and may be of independent interest.

**Theorem 3.3.** *Let $\delta \in (0,1)$ and set $\beta_n = \sqrt{(4/n)((d+1)\log 2n + \log(8/\delta))}$. Under assumption (A1), with probability at least $1 - \delta$ we have that*

$$\sup_x |\hat{r}_p(x) - r_p(x)| \leq C(\beta_n^2 + \beta_n \sqrt{p}),$$

*where $C$ is the constant introduced in Assumption (A1), simultaneously over all $p \in (0,1)$ such that*

$$p + \beta_n^2 + \beta_n \sqrt{p} \leq 1 \quad and \quad p - \beta_n^2 - \beta_n \sqrt{p} \geq 0. \tag{2}$$

The dimension $d$ enters in the previous bound in such a way that, for all $p$ satisfying (2), $\sup_x |\hat{r}_p(x) - r_p(x)| \to 0$ with probability tending to 1 provided that $\frac{d}{n} \to 0$. If we limit the supremum only to the sample points, then the dependence on the dimension disappears altogether and we can instead achieve a nearly-parametric rate of $\sqrt{\frac{\log n}{n}}$.

**Theorem 3.4.** *Let $\delta \in (0,1)$ and set $\alpha_n = \sqrt{(4/(n-1))(\log 2(n-1) + \log(8n/\delta))}$. Under assumption (A1), with probability at least $1 - \delta$ we have that*

$$\max_{i=1,\ldots,n} |\hat{r}_p(X_i) - r_p(X_i)| \leq C(\alpha_n^2 + \alpha_n \sqrt{p} + \frac{2}{n})$$

*where $C$ is the constant introduced in Assumption (A1), simultaneously over all $p \in (0,1)$ such that*

$$p + \alpha_n^2 + \alpha_n \sqrt{p} \leq 1 \quad and \quad p - 2/n - \alpha_n^2 - \alpha_n \sqrt{p - 2/n} \geq 0. \tag{3}$$

The results in Theorem 3.3 and Theorem 3.4 yield the following uniform bounds for the DTM of all order.

**Theorem 3.5.** *Under assumption (A0) and (A1), with probability at least* $1 - \delta$,

$$\sup_x |d(x) - \hat{d}(x)| \leq C_1 \beta_n (\beta_n + \sqrt{m}), \tag{4}$$

*and*

$$\max_{i=1,\ldots,n} |d(X_i) - \hat{d}(X_i)| \leq C_2 \alpha_n (\alpha_n + \sqrt{m} + \frac{2}{n}). \tag{5}$$

*where* $\beta_n$ *and* $\alpha_n$ *are defined in Theorem 3.3 and Theorem 3.4 and* $C_1$ *and* $C_2$ *are some positive constants depending on q, the diameter bound L in Assumption (A0) and the constant C in Assumption (A1).*

**Remark.** *The bound in Theorem 3.5 holds for all choices of* $q \geq 1$*, including the case of* $q = \infty$. *Evaluating explicitly the integral* $\int_0^m (\beta_n + \sqrt{p})^q \, dp$ *will bring out an explicit dependence on q but will not lead to better rates.*

### 3.3 DTM for anomaly detection: theoretical guarantees

We are now ready to derive some theoretical guarantees on the performance of DTM-based methods for discriminating normal and anomalous points in the sample $(X_1, \ldots, X_n)$ according to the Huber-contamination model described above in Section 3.1. We recall that in our setting, a sample point $X_i$ is normal if it belongs to the support $S_0$ of $P_0$, and is otherwise deemed an anomaly if it lies in $S_1$, the support of $P_1$, where $S_1 \cap S_0 = \emptyset$.

The methodology we consider is quite simple, and it is consistent with the prevailing practice of assigning to each sample point a score that expresses its degree of being anomalous compared to the other points. In detail, we rank the sample points based on their empirical DTM values, and we declare the points with largest empirical DTM values as anomalies. This simple procedure will work perfectly well if

$$\max_{X_i \in S_0} \hat{d}(X_i) < \min_{X_i \in S_1} \hat{d}(X_i)$$

and if the difference between the two quantities is large. In general, of course, one would expect that some sample points in $S_0$ may have smaller empirical DTMs of some of the points in $S_1$. The extent to which such incorrect labeling occurs depends on two key factors: how closely the empirical DTM tracks the true DTM and whether the population DTM could itself discriminate normal points versus anomalous ones. The former issue can be handled using the high probability bounds on the stochastic fluctuations of the empirical DTM obtained in the previous section. The latter issue will instead require to specify some degree of separation between the mixture components $P_0$ and $P_1$, both in terms of the distance between their supports but also in terms of how their probability mass gets distributed. There is more than one way to formalize this setting. Here we choose to remain completely agnostic to the form of the contaminating distribution $P_1$, for which we impose virtually no constraint. On the other hand, we require the normal distribution $P_0$ to satisfy condition (A2) above in such a way that point inside the support will have larger values of $a(x)$ than points near the boundary of $S_0$. This condition, which is satisfied if for example $P_0$ admits a Lebesgue density whose values increase as a function of the distance from the boundary of $S_0$, ensures that the population DTM will be large near the boundary of $S_0$ and small everywhere else. As a result, incorrect labeling of normal points will only occur around the boundary of $S_0$ but not inside the bulk of the distribution $P_0$. We formalize this intuition in our next result, which is purely deterministic.

**Proposition 3.6.** *Under assumptions (A0) and (A2), suppose that* $a(x) = g(d(x, \partial S_0))$*, where* $g(z)$ *is a non-decreasing function on* $[0, z_0)$ *for some* $z_0$*, and* $g(z) \geq g(z_0)$ *for all* $z \geq z_0$*. Let*

$$\eta = \min_{x \in S_0, y \in S_1} \|x - y\| \tag{6}$$

*be the distance between* $S_0$ *and* $S_1$ *and* $h > 0$ *be a given threshold parameter. For any* $m > \varepsilon$*, additionally assume that*

$$g(z_0) \geq g_0 := \begin{cases} \frac{m}{1-\varepsilon} \left( \frac{b+q}{b} \left( \frac{m-\varepsilon}{m} \eta^q - h \right) \right)^{-b/q} & 1 \leq q < \infty \\ \frac{m}{1-\varepsilon} (\eta - h)^{-b} & q = \infty. \end{cases} \tag{7}$$

*Next, define the "safety zone" $A_\eta$ as*

$$A_\eta = \left\{ x \in S_0 : d(x, \partial S_0) \geq g^{-1}(g_0) \right\} \tag{8}$$

*Then, we have*

$$\sup_{x \in A_\eta} d_{P,m,q}(x) + h < \inf_{y \in S_1} d_{P,m,q}(y). \tag{9}$$

The main message from the previous result is that there exists a subset $A_\eta$ of the support of the normal distribution, which intuitively corresponds to a region deep inside the support of $P_0$ of high density, over which the population DTM will be smaller than at any point in the support $S_1$ of the contaminating distribution. Thus, the true DTM is guaranteed to perfectly separate $A_\eta$ from $S_1$, making mistakes (possibly) only for the normal points in $S_0 \setminus A_\eta$.

Notice that the definition of $A_\eta$ depends on all the relevant quantities, namely the contamination parameter $\varepsilon$, the probability parameter $m$, the dimension $b$ of $P_0$ and the order $q$ of the DTM through the expression (7). Importantly, it is necessary that $m > \varepsilon$, otherwise inequality (9) maybe not be satisfied. For example, we can take $P_1$ to have point mass at a single point $y$; then $r_{P,t}(y) = 0$ for all $t \leq m$, and the right hand side of (9) is zero.

When $g(0) = a_0 > 0$, which occurs, e.g., if $P_0$ has a density bounded away from $0$ over its support, implies that $A_\eta = S_0$ if

$$\eta > \left( \frac{m}{m - \varepsilon} \left( \frac{b}{b + q} \left( \frac{m}{a_0(1 - \varepsilon)} \right)^{q/b} + h \right) \right)^{-1/q}.$$

That is, when $S_0$ and $S_1$ are sufficiently well-separated, the DTM will classify all the points in $S_0$ as normals.

The parameter $h$ serves as a buffer that allows one to replace the DTM function $d(x)$ with any estimator that is close to it in the supremum norm by no more than $h$. Thus, we may plug-in the high-probability bounds of Theorem 3.4 and Theorem 3.3 to conclude that the empirical DTM will will identify all normal instances within $A_\eta$ correctly, with high probability.

**Corollary 3.6.1.** *Taking $h$ to be twice the upper bound in* (5)*, we get, with probability at least $1 - \delta$,*

$$\max_{X_i \in A_\eta} \hat{d}_{P,m,q}(X_i) < \min_{X_i \in S_1} \hat{d}_{P,m,q}(X_i).$$

*Similarly, if $h$ is twice the upper bound in* (4)*, we have that*

$$\sup_{x \in A_\eta} \hat{d}_{P,m,q}(x) < \inf_{y \in S_1} \hat{d}_{P,m,q}(y). \tag{10}$$

The guarantee in (10) calls for a higher sample complexity that depends on the dimension $d$. At the same time, it extends to all the points in $A_\eta$ and not just the sample points. Thus the DTM can accurately identify not only the normal instance in the sample but any other normal instance, such as future observations.

## 3.4 Illustrative examples

We illustrate the separation condition in Proposition 3.6 with the following example. Consider a collection of normal instances generated from a standard normal distribution. Figure 2 shows the mis-classification rates for $\mathrm{DTM}_2$ as a cluster of 5 anomalies approaches the normal instances for three different underlying distributions: Gaussian, Moon-shaped, Circle. The color of each point represents its class, with black being the normal instances and red being anomalies. The radius of the circle around each point represents its empirical DTM score, and the color of the circle represents its predicted class from $\mathrm{DTM}_2$. As we see, as the anomalies approach the normal instances, more and more data around the boundaries of the normal distribution get mis-classified as anomalies.

## 4 Conclusions

In this paper we have presented empirical evidence, based on simulated and real-life benchmark datasets, that NN-based methods show very good performance at identifying anomalous instances

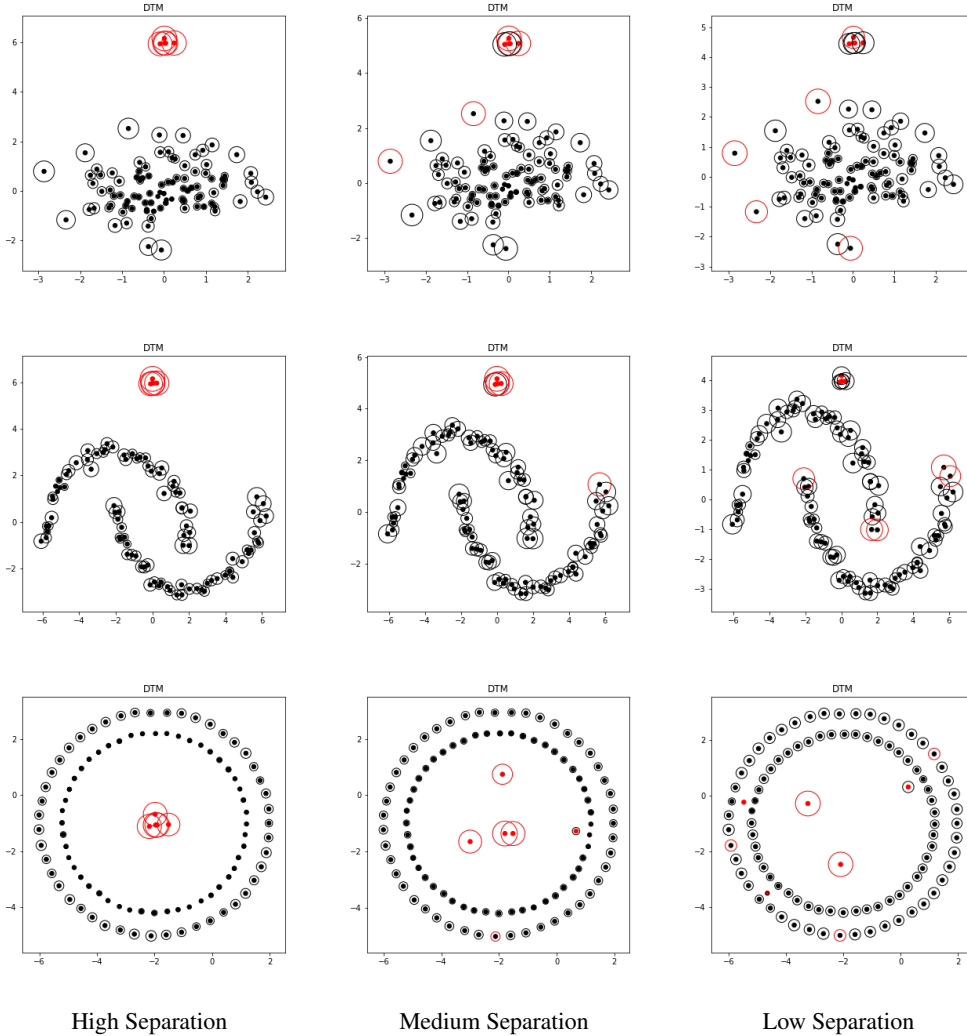

High Separation               Medium Separation               Low Separation

Figure 2: Performance of DTM when the separation distance between the normal instances and anomalies gradually decreases. Top: Gaussian; Middle: Moon-Shaped; Bottom: Circle.

in an unsupervised anomaly detection set-up. We have introduced a simple but natural framework for anomaly detection based on the Huber contamination model and have used it to characterize the performance of a class of NN methods for anomaly detection that are based on the distance-to-a-measure (DTM) functional. In our results we rely on various geometric and analytic properties of the underlying distribution to the accuracy of DTM-methods for anomaly detection. We are able to demonstrate that, under mild conditions, NN methods will mis-classify normal points only around the boundary of the support of the distribution generating normal instances and have quantified this phenomenon rigorously. Finally, we have derived novel finite sample bounds on the nearest neighbor radii and on the rate of convergence of the empirical DTM to the true DTM that may be of independent interest.

## Footnotes

[1]$\text{DTM}_2$ stands for the empirical DTM (see Section 3) with $q = 2$. We include its empirical analysis here for comparison purposes.

[2]`https://github.com/xgu1/DTM`

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
