[Supplementary Material]

# Appendices

## A    Definition for $\mathrm{DTMF}_2$

**Definition A.1.** *The* $\mathrm{DTM}_2$ *and* $\mathrm{DTMF}_2$ *scores are defined as:*

$$\mathrm{DTM}_2(x) = \left( \frac{1}{k} \sum_{X_i \in N_k(x)} \|X_i - x\|^2 \right)^{1/2},$$

$$\mathrm{DTMF}_2(x) = \frac{1}{|N_k(x)|} \sum_{y \in N_k(x)} \frac{\mathrm{DTM}_2(y)}{\mathrm{DTM}_2(x)}.$$

## B    Proof of Theorems

### B.1    Proof of Theorem 3.3

*Proof.* By standard VC theory [1], for any ball $B \subset \mathbb{R}^d$, we have

$$P(B) \geq p + \beta_n^2 + \beta_n \sqrt{p} \Rightarrow P_n(B) \geq p. \tag{1}$$

$$P(B) < p - \beta_n^2 - \beta_n \sqrt{p} \Rightarrow P_n(B) < p \tag{2}$$

with probability at least $1 - \delta$.

*Step1:* First, we want to show that

$$\hat{r}_p(x) \leq r_p(x) + C(\beta_n^2 + \beta_n \sqrt{p}) \tag{3}$$

for all $x$. By definition of $r_p(x)$, we have $P(B(x, r_p(x))) \geq p$. Define $r^+ = \inf\{r : P(B(x, r)) \geq p + \beta_n^2 + \beta_n \sqrt{p}\}$. Then, we have

$$P(B(x, r^+)) \geq p + \beta_n^2 + \beta_n \sqrt{p} \Rightarrow P_n(B(x, r^+)) \geq p$$

by (1). Therefore, $r^+ \geq \hat{r}_p(x)$. Next, note that $r_p(x) \leq r^+$. If $r_p(x) = r^+$, (3) holds trivially. If $r_p(x) < r^+$, then for all $s$ such that $r_p(x) < s < r^+$, we have

$$p \leq P(B(x, r_p(x))) \leq P(B(x, s)) \leq p + \beta_n^2 + \beta_n \sqrt{p}.$$

Then by assumption (A1),

$$s \leq r_p(x) + C(\beta_n^2 + \beta_n \sqrt{p}).$$

Taking $s \uparrow r^+$, we get $\hat{r}_p(x) \leq r^+ \leq r_p(x) + C(\beta_n^2 + \beta_n \sqrt{p})$ as desired.

*Step 2:* Next, we want to show the reverse direction:

$$r_p(x) \leq \hat{r}_p(x) + C(\beta_n^2 + \beta_n \sqrt{p}). \tag{4}$$

Let $r^- = \inf\{r : P(B(x, r)) \geq p - \beta_n^2 - \beta_n \sqrt{p}\}$. Then, clearly $r^- \leq r_p(x)$ and $P(B(x, r^-)) \geq p - \beta_n^2 - \beta_n \sqrt{p}$. For all $s < r^-$, we have

$$P(B(x, s)) < p - \beta_n^2 - \beta_n \sqrt{p}$$
$$\Rightarrow P_n(B(x, s)) < p$$
$$\Rightarrow s < \hat{r}_p(x)$$

where the first implication follows from (2). Taking $s \uparrow r^-$, we get $r^- \leq \hat{r}_p(x)$. If $r^- = r_p(x)$, (4) holds trivially. If $r^- < r_p(x)$, then for any $u$ satisfying $r^- < u < r_p(x)$, we have

$$p - \beta_n^2 - \beta_n \sqrt{p} \leq P(B(x, r^-)) \leq P(B(x, u)) \leq p$$
$$\Rightarrow u \leq r^- + C(\beta_n^2 + \beta_n \sqrt{p}).$$

Taking $u \uparrow r_p(x)$, we get

$$r_p(x) \leq r^- + C(\beta_n^2 + \beta_n \sqrt{p}) \leq \hat{r}_p(x) + C(\beta_n^2 + \beta_n \sqrt{p})$$

as desired. $\qquad\square$

## B.2 Proof of Theorem 3.4

*Proof.* Fix $X_i$, and let $P_{i,n-1}$ be the marginal distribution of $\mathbb{X}_n \backslash \{X_i\}$. Then by standard VC theory [1], for any ball $B \subset \mathbb{R}^d$ with fixed center $X_i$ and arbitrary radius, we have

$$P(B) \geq p + \alpha_n^2 + \alpha_n \sqrt{p} \Rightarrow P_{i,n-1}(B) \geq p. \tag{5}$$

$$P(B) < p - \alpha_n^2 - \alpha_n \sqrt{p} \Rightarrow P_{i,n-1}(B) < p \tag{6}$$

with probability at least $1 - \delta/n$. Define $\hat{r}_p^{-i}(X_i)$ to be the $p$-NN radius associated with the empirical measure $P_{i,n-1}$.

*Step 1*: Following the same steps as in Step 1 in the proof of Theorem 3.3, we have

$$\hat{r}_p^{-i}(X_i) \leq r_p(X_i) + C(\alpha_n^2 + \alpha_n \sqrt{p}). \tag{7}$$

*Step 2*: Next, we want to show the following:

$$r_p(X_i) \leq \hat{r}_{p'}^{-i}(X_i) + C(\alpha_n^2 + \alpha_n \sqrt{p} + \frac{2}{n}), \tag{8}$$

where $p' = p - 2/n$. Following the same steps as in step 2 in the proof of Theorem 3.3, let $r^- = \inf\{r : P(B(X_i, r)) \geq p' - \alpha_n^2 - \alpha_n \sqrt{p'}\}$. Then, clearly $r^- \leq r_{p'}(X_i) \leq r_p(X_i)$ and $P(B(X_i, r^-)) \geq p' - \alpha_n^2 - \alpha_n \sqrt{p'}$. For all $s < r^-$, we have

$$P(B(X_i, s)) < p' - \alpha_n^2 - \alpha_n \sqrt{p'}$$
$$\Rightarrow P_{i,n-1}(B(X_i, s)) < p'$$
$$\Rightarrow s < \hat{r}_{p'}^{-i}(X_i)$$

where the first implication follows from (6). Taking $s \uparrow r^-$, we get $r^- \leq \hat{r}_{p'}^{-i}(X_i)$. If $r^- = r_p(X_i)$, (8) holds trivially. If $r^- < r_p(X_i)$, then for any $u$ satisfying $r^- < u < r_p(X_i)$, we have

$$p' - \alpha_n^2 - \alpha_n \sqrt{p'} \leq P(B(X_i, r^-)) \leq P(B(X_i, u)) \leq p$$
$$\Rightarrow u \leq r^- + C(\alpha_n^2 + \alpha_n \sqrt{p'} + \frac{2}{n}).$$

Taking $u \uparrow r_p(X_i)$, we get

$$r_p(X_i) \leq r^- + C(\alpha_n^2 + \alpha_n \sqrt{p'} + \frac{2}{n}) \leq \hat{r}_{p'}^{-i}(X_i) + C(\alpha_n^2 + \alpha_n \sqrt{p} + \frac{2}{n})$$

as desired.

*Step 3*: Finally, we want to show that

$$\hat{r}_{p'}^{-i}(X_i) \leq \hat{r}_p(X_i) \leq \hat{r}_p^{-i}(X_i). \tag{9}$$

Let $k = \lceil (n-1)p \rceil$, then we have $p \leq \frac{k}{n-1}$. By construction,

$$P_{i,n-1}(B(X_i, \hat{r}_p^{-i}(X_i))) = \frac{k}{n-1} \implies P_n(B(X_i, \hat{r}_p^{-i}(X_i))) = \frac{k+1}{n} \geq \frac{k}{n-1} \geq p,$$

which implies that $\hat{r}_p(X_i) \leq r_p^{-i}(X_i)$. Similarly, let $k = \lceil (n-1)p' \rceil$, then we have $p' \geq \frac{k-1}{n-1}$. By construction,

$$P_{i,n-1}(B(X_i, \hat{r}_{p'}^{-i}(X_i))) = \frac{k}{n-1} \implies P_n(B(X_i, \hat{r}_{p'}^{-i}(X_i))) = \frac{k+1}{n} \leq \frac{k-1}{n-1} + \frac{2}{n} \leq p,$$

which implies that $\hat{r}_{p'}^{-i}(X_i) \leq \hat{r}_p(X_i)$.

Combining the results from Step 1,2,3, we get

$$|r_p(X_i) - \hat{r}_p(X_i)| \leq C(\alpha_n^2 + \alpha_n \sqrt{p} + \frac{2}{n})$$

with probability at least $1 - \delta/n$. Thus,

$$\max_{i=1,\ldots,n} |r_p(X_i) - \hat{r}_p(X_i)| \leq C(\alpha_n^2 + \alpha_n \sqrt{p} + \frac{2}{n})$$

with probability at least $1 - \delta$ by union bound. $\qquad \square$

## B.3 Proof of Theorem 3.5

*Proof.* Here, we include the proof for (**??**), the proof for (**??**) can be argued in a similar fashion. The conditions $p + \beta_n^2 + \beta_n\sqrt{p} \leq 1$ and $p - \beta_n^2 - \beta_n\sqrt{p} \geq 0$ imply $c_1\beta_n^2 \leq p \leq 1 - c_2\beta_n^2$ for some constant $c_1$ and $c_2$. Let $I_1 = \{p : c_1\beta_n^2 \leq p \leq 1 - c_2\beta_n^2\} \cap [0, m]$ and $I_2 = [0, m]\backslash I_1$. Then, with probability at least $1 - \delta$, for $p \in I_1$, $\sup_x |r_p(x) - \hat{r}_p(x)| \leq C\beta_n(\beta_n + \sqrt{p})$ by Theorem 3.3. For $p \in I_2$, $\sup_x |r_p(x) - \hat{r}_p(x)| \leq 2L$ where $L$ is the bound on the support of $P$ by Assumption (A0).

$$
\begin{aligned}
\sup_x |d(x) - \hat{d}(x)| = \sup_x &\left| \left( \frac{1}{m} \int_0^m r_p(x)^q \, dp \right)^{1/q} - \left( \frac{1}{m} \int_0^m \hat{r}_p(x)^q \, dp \right)^{1/q} \right| \\
\leq &\sup_x \left( \frac{1}{m} \int_0^m (r_p(x) - \hat{r}_p(x))^q \, dp \right)^{1/q} \\
\leq &\left( \frac{1}{m} \int_{I_1} (C\beta_n(\beta_n + \sqrt{p}))^q \, dp + \frac{1}{m} \int_{I_2} (2L)^q \, dp \right)^{1/q} \\
\leq &\left( (C\beta_n(\beta_n + \sqrt{m}))^q \frac{m - c_1\beta_n^2}{m} + (2L)^q \frac{c_1\beta_n^2 + m - m \wedge (1 - c_2\beta_n^2)}{m} \right)^{1/q} \\
\leq &\, C_1\beta_n(\beta_n + \sqrt{m})
\end{aligned}
$$

for some positive number $C_1$, dependent on $q$. $\qquad\square$

## B.4 Proof of Proposition 3.6

*Proof.* Equivalently, by the definition of DTM, we will need to show that

$$
\sup_{x \in A_\eta} \frac{1}{m} \int_0^m r_{P,t}(x)^q \, dt < \inf_{y \in S_1} \frac{1}{m} \int_0^m r_{P,t}(y)^q \, dt - h. \tag{10}
$$

Since for any $x \in A_\eta$, $P(B(x,r)) \geq (1 - \varepsilon)P_0(B(x,r))$, we have $r_{P,t}(x) \leq r_{(1-\varepsilon)P_0,t}(x) = r_{P_0,t/1-\varepsilon}(x)$. When $r = (\frac{t}{a(x)(1-\varepsilon)})^{1/b}$, by the $(a,b)$-condition of $P_0$, we have $P_0(B(x,r)) \geq a(x)r^b = \frac{t}{1-\varepsilon}$. Hence, $r_{P_0,t/1-\varepsilon}(x) \leq (\frac{t}{a(x)(1-\varepsilon)})^{1/b}$. Putting the inequalities together, the LHS of (10) gives

$$
\begin{aligned}
\frac{1}{m} \int_0^m r_{P,t}(x)^q \, dt &\leq \frac{1}{m} \int_0^m \left( \frac{t}{a(x)(1-\varepsilon)} \right)^{q/b} dt \\
&= \frac{b}{b+q} \left( \frac{m}{a(x)(1-\varepsilon)} \right)^{q/b}
\end{aligned}
$$

Next, consider the right hand side of (10). We have that

$$
\begin{aligned}
\inf_{y \in S_1} \frac{1}{m} \int_0^m r_{P,t}(y)^q \, dt &= \inf_{y \in S_1} \frac{1}{m} \int_0^\varepsilon r_{P,t}(y)^q \, dt + \frac{1}{m} \int_\varepsilon^m r_{P,t}(y)^q \, dt \\
&\geq 0 + \frac{1}{m} \int_\varepsilon^m \eta^q \, dt \\
&= \frac{m - \varepsilon}{m} \eta^q
\end{aligned}
$$

Hence, (10) holds if

$$
\frac{b}{b+q} \left( \frac{m}{a(x)(1-\varepsilon)} \right)^{q/b} < \frac{m - \varepsilon}{m} \eta^q - h \tag{11}
$$

$$
\Leftrightarrow a(x) > \frac{m}{1-\varepsilon} \left( \frac{b+q}{b} \left( \frac{m-\varepsilon}{m}\eta^q - h \right) \right)^{-b/q} \tag{12}
$$

$$
\Leftrightarrow x \in A_\eta \tag{13}
$$

$\qquad\square$

|            |                    |                        |
|------------|--------------------|------------------------|
| (a) Ring   | (b) Local Anomalies | (c) Clustered Anomalies |

Figure 1: Examples of difficult datasets.

## C   Simulation Results on 23 Real Datasets from ODDS

Table 1 gives the exact AUC and AP scores of IForest, LODA, LOF, $DTM_2$, $k$NN, and $k^{th}$NN on 23 real datasets from the ODDS library.

## D   Performance on Difficult Examples

Figure 1 gives three examples of difficult situations where some algorithms will very likely fail. The black dots represent the normal instances, and the two red dots represent anomalies. In Figure 1a where the anomalies are located in the center of a circle of normal points, IForest and LODA will have a hard time detecting the anomalies, whereas LOF and NN-methods have no trouble. In Figure 1b, if the anomalies are locally relatively far away from a group of normal points, NN-methods, IForest, and LODA won't be able to pick them up, whereas LOF is designed to handle this specific case. However, we observed through extensive simulations that LOF can easily make mistakes on global anomalies, and Figure 1c gives one such example. If we have a cluster of anomalies located at some distance from a collection of normal points, LOF tends to mis-identify some of the anomalies as normal points, whereas the other methods have no such problem.

Figure 2 3 4 show the performance of IForest, LODA, LOF, and $DTM_2$ in each of the difficult examples. The radius of the circle around each point gives the anomaly score of each algorithm, and the color of the circle represents the predicted class by the algorithm.

## References

[1] Kamalika Chaudhuri, Sanjoy Dasgupta, Samory Kpotufe, and Ulrike von Luxburg. Consistent procedures for cluster tree estimation and pruning. *IEEE Transactions on Information Theory*, 60:7900–7912, 2014.

Table 1: Performance of IForest, LODA, LOF, DTM$_2$, $k$NN, and $k^{\text{th}}$NN on 23 real datasets from the ODDS library.

| AUC | IForest | LODA | LOF | DTM$_2$ | $k$NN | $k^{\text{th}}$NN |
|---|---|---|---|---|---|---|
| annthyroid | 0.846217 | 0.711716 | 0.688763 | 0.677126 | 0.681196 | 0.662250 |
| arrhythmia | 0.774180 | 0.789645 | 0.763778 | 0.807466 | 0.806681 | 0.815473 |
| breastw | 0.988089 | 0.987891 | 0.376371 | 0.980041 | 0.979805 | 0.982081 |
| cardio | 0.925666 | 0.904219 | 0.705637 | 0.831097 | 0.820695 | 0.880306 |
| glass | 0.706775 | 0.771816 | 0.737669 | 0.867751 | 0.867209 | 0.869106 |
| ionosphere | 0.842363 | 0.853369 | 0.899506 | 0.928007 | 0.928148 | 0.920141 |
| letter | 0.600280 | 0.622487 | 0.842000 | 0.856193 | 0.861893 | 0.809837 |
| lympho | 1.000000 | 0.992958 | 0.981221 | 0.977700 | 0.977700 | 0.978286 |
| mammography | 0.853864 | 0.866368 | 0.819344 | 0.850100 | 0.850604 | 0.849169 |
| mnist | 0.792829 | 0.595506 | 0.839678 | 0.862295 | 0.861369 | 0.861813 |
| musk | 0.999944 | 0.994193 | 0.286222 | 0.957031 | 0.936976 | 1.000000 |
| optdigits | 0.714978 | 0.714282 | 0.612373 | 0.560559 | 0.537313 | 0.842404 |
| pendigits | 0.961689 | 0.950902 | 0.850733 | 0.958278 | 0.950210 | 0.970528 |
| pima | 0.675037 | 0.618657 | 0.557993 | 0.636045 | 0.634418 | 0.639545 |
| satellite | 0.686132 | 0.725766 | 0.578879 | 0.768331 | 0.764688 | 0.795738 |
| satimage-2 | 0.993326 | 0.994631 | 0.991675 | 0.999054 | 0.999079 | 0.998954 |
| shuttle | 0.997529 | 0.992264 | 0.522135 | 0.989215 | 0.984996 | 0.993954 |
| speech | 0.441678 | 0.441248 | 0.478689 | 0.482781 | 0.483310 | 0.478594 |
| thyroid | 0.978939 | 0.954587 | 0.963042 | 0.946970 | 0.947420 | 0.943083 |
| vertebral | 0.359048 | 0.338889 | 0.495714 | 0.331746 | 0.330794 | 0.323968 |
| vowels | 0.739488 | 0.757411 | 0.937155 | 0.961067 | 0.963144 | 0.946216 |
| wbc | 0.943177 | 0.958517 | 0.910764 | 0.948113 | 0.946379 | 0.949980 |
| wine | 0.776471 | 0.963025 | 0.428151 | 0.994958 | 0.993277 | 0.996218 |
| avg.rank | 4.086957 | 4.478261 | 6.391304 | 4.086957 | 4.347826 | 3.565217 |

(a) AUC

| AUC | IForest | LODA | LOF | DTM$_2$ | $k$NN | $k^{\text{th}}$NN |
|---|---|---|---|---|---|---|
| annthyroid | 0.336719 | 0.221278 | 0.252282 | 0.201405 | 0.203313 | 0.191132 |
| arrhythmia | 0.422741 | 0.479021 | 0.372709 | 0.491718 | 0.489785 | 0.511596 |
| breastw | 0.972689 | 0.970735 | 0.272824 | 0.945230 | 0.944475 | 0.951773 |
| cardio | 0.577570 | 0.579294 | 0.202455 | 0.404516 | 0.399020 | 0.450174 |
| glass | 0.096007 | 0.140315 | 0.193538 | 0.162208 | 0.162824 | 0.155266 |
| ionosphere | 0.794018 | 0.794706 | 0.866694 | 0.928604 | 0.928993 | 0.911973 |
| letter | 0.089123 | 0.090754 | 0.334208 | 0.260399 | 0.268795 | 0.200453 |
| lympho | 1.000000 | 0.835714 | 0.655556 | 0.723611 | 0.723611 | 0.695202 |
| nmammography | 0.193517 | 0.264330 | 0.130258 | 0.167475 | 0.169236 | 0.161568 |
| mnist | 0.267129 | 0.143844 | 0.397000 | 0.404172 | 0.403502 | 0.387391 |
| musk | 0.998328 | 0.881940 | 0.021850 | 0.618577 | 0.496054 | 1.000000 |
| noptdigits | 0.051332 | 0.047997 | 0.033582 | 0.032489 | 0.031128 | 0.081907 |
| npendigits | 0.328479 | 0.263207 | 0.077311 | 0.217698 | 0.193527 | 0.315036 |
| npima | 0.506879 | 0.491468 | 0.391361 | 0.486558 | 0.485157 | 0.492184 |
| satellite | 0.659824 | 0.693257 | 0.406871 | 0.639164 | 0.634576 | 0.680913 |
| satimage-2 | 0.936035 | 0.911899 | 0.516222 | 0.972246 | 0.972113 | 0.972834 |
| shuttle | 0.983694 | 0.825359 | 0.261309 | 0.746971 | 0.694083 | 0.818767 |
| speech | 0.016421 | 0.015298 | 0.018916 | 0.019062 | 0.019088 | 0.018781 |
| thyroid | 0.595528 | 0.276667 | 0.397719 | 0.297644 | 0.296979 | 0.285007 |
| vertebral | 0.094209 | 0.090683 | 0.121829 | 0.089739 | 0.089664 | 0.088901 |
| vowels | 0.179951 | 0.181478 | 0.396334 | 0.484752 | 0.501906 | 0.403366 |
| wbc | 0.588631 | 0.640832 | 0.279934 | 0.495254 | 0.488687 | 0.554438 |
| wine | 0.192461 | 0.544417 | 0.072027 | 0.941540 | 0.928312 | 0.954040 |
| avg.rank | 3.521739 | 4.260870 | 6.260870 | 4.347826 | 4.739130 | 3.869565 |

(b) AP

(a) IForest

(b) LODA

(c) $DTM_2$

(d) LOF

Figure 2: Performance on the difficult datasets. Case: ring

(a) IForest

(b) LODA

(c) DTM$_2$

(d) LOF

Figure 3: Performance on the difficult datasets. Case: local anomalies

(a) IForest

(b) LODA

(c) DTM$_2$

(d) LOF

Figure 4: Performance on the difficult datasets. Case: clustered anomalies