[Reviews · NeurIPS 2019]

Reviewer 1



Update after author response --------------------------------- I would like to thank the authors for their response. "most p" is still somewhat not very precise and I would like the author to clarify the issues I raised in the final version. Summary of the main ideas: ---------------------------------- The authors theoretically study Nearest Neighbor methods for anomaly detection in the unsupervised setting using the distance-to-a-measure (DTM) quantity. If I understood correctly the major differences from the related work are - [1, 2, 3] (papers about theoretical analyses of kNN methods for anomaly detection) consider the semi-supervised setting (novelty detection) whereas this paper deals with the unsupervised setting - [1, 2, 3] use different techniques (geometric entropy and conformal prediction) and do not rely on the general DTM quantity. - the DTM related work gives a limiting distribution and confidence sets but no finite sample bounds. The authors start the paper by an extensive benchmark showing that Nearest Neighbor methods perform reasonably well compared to other state-of-the-art methods. They then prove finite sample bounds on the deviation of the empirical k-th Nearest neighbor radius and the empirical distance-to-a-measure (DTM) from their true counterparts. They finally apply these results to anomaly detection using kNN methods. Originality ------------ This paper is the first, according to the authors and the cited related work, to state finite sample bounds on the deviation of the empirical k-th Nearest neighbor radius and the empirical distance-to-a-measure (DTM) from their true counterparts and to apply these results to anomaly detection. Quality and clarity ---------------------- The paper is well-written and easy to read. Having a second section about experiments was a bit unusual but this motivates the analysis of the next sections. However I would add a sentence in the introduction of section 2 to emphasize that the methods/algorithms concerned by the theoretical analysis of this paper are k-NN, kth-NN and DTM2. It could be useful to restate at the end of section 3.1 what is mentioned in the introduction of the paper about the difference with the theoretical studies done in [1, 2, 3]. This would clarify a bit more the difference with the related work when reading the theoretical section. The authors should also clarify the main differences (listed in the Summary, if correct). In section 2, for the experiments of Figure 1, hyperparameters for kNN, kthNN and DTM2 methods are clearly stated (k = 0.03 * sample size). I assume the same choice is made for LOF (I would make this clearer). Hyperparameters for LODA and Iforest are absent (e.g. number of trees and subsampling parameter for Iforest). In section 3.4, we do not know which threshold the authors are using to declare a sample as abnormal or not. I liked the explanations and interpretations of the assumptions/results/definitions given in the paper. I found them very clear and useful. I would like the authors to clarify where in [14] we can find Assumption A1. Some notations are used at different places in the paper with different meanings: - epsilon : epsilon in [0, 1) is used in the contamination model (~ l.123) to denote the proportion of contamination. it is also used in proposition 3.6 with the same meaning. However it is used in the definition of Assumption A1 with a different meaning. - p : the authors should be more careful when using p. First it is said l.134 that for the rest of the article p = k/n but this is not true everywhere in the paper: for instance it is used in equation (1) as the variable of integration, and for equation (2) in Theorem 3.5 to be true we need Theorem 3.3 to be true for all p in [0, 1), not just k/n. The authors need to better clarify when p = k/n (e.g. in the interpretation of \hat r_p(x) as the kth-NN radius or in Theorem 3.4) or when p is/needs to be an arbitrary number in [0, 1). - eta : eta is used in the definition of assumption A1 and in equation (14), both having a different meaning. These inconsistencies should be fixed for a better clarity and to prevent the reader from being confused. About Theorem 3.5, equation (2) : for the proof to hold, Theorem 3.3 must be true for all p in [0,1), as the integral over all p in [0, m) is considered, m in [0, 1). Note also that the event with probability greater than 1 - delta such that the inequality of Theorem 3.3 is true shall not depend on p, otherwise additional arguments are needed to ensure that we can integrate over [0, m) and still have the result with probability at least 1 - delta ([0, m) contains uncountably many p). The fact that the event does not depend on p is not clear in the paper. Although this can easily be checked in the proof, it would be much better if the authors make this clear when stating the result, e.g. "with probability at least 1 - delta: for all p in [0, 1), ..." and maybe add a sentence to say that the event does not depend on p. About the proof of Theorem 3.3 : the authors could also clarify that this holds for all p in [0, 1) by writing something like "with probability at least 1 - delta: for any ball B and for all p in [0, 1)", especially because as explained above we need the result to be true for all p and the event to not depend on p but also because in the reference given by the authors, equations (1) and (2) are stated only for p=k/n. They could also add in the supplementary material that these results are Lemma C.2 and Theorem C.1 of [2], except that these results are here stated for all p instead of p=k/n. Besides, the needed reference is [2] and not [1]. [1] is stated in [2]. I would consider removing [1] here or better explain how these inequalities are obtained: e.g. "equations (1) and (2) are proven in [2] by standard VC theory". About Theorem 3.5, equation (3) : I would need additional arguments/explanations to be convinced that this result holds. No proof is given. I assumed that I could follow the proof of equation (2) but Theorem 3.4 only holds for p=k/n and we need it to hold for all p as explained above. The authors can maybe use the fact that \hat r_p(x) is piecewise constant as a function of p? Minor comments: - l.120 : n i.i.d. realization -> n i.i.d. realizations - l.126 : our goal is devise -> to devise - l.127 : anomalous one -> anomalous ones - l.137 and l.140 (in assumptions (A1) and (A2)) : There exists -> There exist - equation following l.205 -> it would be better to use the same notation under the min as the one used under the max. Besides x_I is undefined. - l.213 : in terms the distance -> in terms of the distance - l.221 : the bulk the distribution -> the bulk of the distribution - The authors wrote in the paper (l.62) that the code for the experiments will be made publicly available so I assume that this means once (and if) the paper is accepted while in the reproducibility checklist they said that they provided a link to a downloadable source code. Significance ------------ The finite sample bounds derived in this paper are likely to be of interest to derive new theoretical results. NN methods for anomaly detection are already famous among practitioners and it is important to have theoretical guarantees.

Reviewer 2



The main contributions of this paper are described in the previous section. The presented empirical study follows the steps of others (as cited by the authors), but studying the DTM-based models is original. The paper is generally well-written, presenting the results clearly. It is somewhat confusing that Table1 shows failure rates (lower is better), while Table2 shows performance (higher is better). The empirical study is comprehensive, and the theoretical analysis is definitely non-trivial. The significance of the paper is mainly in suggesting the use of DTM2 for anomaly detection and pointing out the possible advantage of nearest neighbor based methods for anomaly detection in high dimensional data. While this is probably not a breakthrough in the field, it is of significant interest to data science practitioners. I would grade this paper as a "good submission".

Reviewer 3



This paper proposes a new outlierness measure, called DTM, for outlier detection, and theoretically analyze its performance. In addition, the paper performs experiments on various real-world datasets to examine the effectiveness of outlier detection methods. Although the theoretical analysis of the paper is interesting, I have the following concerns. - This paper provides several theoretical results for NN-based outlier detection, which is an interesting contribution. However, empirical evaluation does not directly support the obtained theoretical results. For example, how are the assumptions (A0-A2) natural in real-world datasets? How is the result in Proposition 3.6 realistic? - An important empirical result is that an ensemble method is the most effective. However, theoretical comparison between an ensemble method and the proposed method is not performed. - The proposed method includes several parameters. However, the sensitivity is not examined, which is crucial to show the usefulness of the proposed method. - Important references about NN-based outlier detection are missing. In particular, [1] and [2] are known to be the state-of-the-art NN-based methods using sampling with rigorous theoretical analyses. Moreover, [3] also performs more detailed theoretical analysis of [1]. Please refer and discuss the relationship between the proposed method and such methods. [1] Sugiyama, M. Borgwardt, K. M.: Rapid Distance-Based Outlier Detection via Sampling, NIPS2013. [2] Wu, M. and Jermaine, C.: Outlier detection by sampling with accuracy guarantees, SIGKDD22006. [3] Ting, K. M., Washio, T., Wells, J. R., Aryal, S., Defying the gravity of learning curve: a characteristic of nearest neighbour anomaly detectors, Machine Learning, 2016. - Important reference [4], which discusses performance for high-dimensional data, is missing. [4] Zimek, A., Schubert, E., Kriegel, H.-P., A survey on unsupervised outlier detection in high-dimensional numerical data, Statistical Analysis and Data Mining, 2012. ========== Update after Author feedback ========== Although I acknowledge the author feedback, some of my concerns (sensitivity analysis and discussion of high-dimensional data) still remain.

Reviewer 4



Anomaly detection defines a very important task in data analysis, i.e., outliers often have high semantic value, see anomalous tissue samples in histological slides that can indicate early states of cancer. Well implemented empirical studies can guide practitioners in empirical studies. The finite sample bounds also provide a theoretical basis for the comparison of nearest neighbour methods for anomaly detection. The theoretical findings are based on the assumption that the support of the normal distribution does not overlap with the outlier distribution. This assumption is often violated and I believe that Huber in his original contribution did not assume it. Often an anomaly distribution might overlap with the tail of the normal distribution. Such a situation should be discussed in the paper since it happens in practice and for these cases theory should provide guidance. originality: The paper follows the line of research in anomaly detection and puts its findings in relation to the literature. The empirical study has moderate originality since the techniques have been used elsewhere in the literature. The finite sample bounds are interesting and look original to me. quality: The presentation of the empirical results should be improved. The box plots of Fig 1 are not very informative since the different tasks have quite different complexity as can be seen from the absolute AP and AUC values. It would be much better to report on the performance differences between two methods. It could be that method A always performed better than method B given a data set. Then box plots of performance differences between two methods on various data sets might be significant while the box plots of both methods on the various data sets are highly overlapping. clarity: I didn't see a clear strategy in the study The empirical work and the theoretical findings are only very loosely connected to the empirical results. significance: I am not aware of other sample bounds for anomaly detection and therefore, I consider the results as significant. However, I might not be aware of some works on sample bounds for the studied scenario. minor comment: l95: "have significantly downgraded" -> "have been significantly downgraded"

[Author Response · NeurIPS 2019]

**Reviewer #1:**

*Writing.* We will fix the typos, notation inconsistencies (e.g. about the parameters $\epsilon$ and $\eta$), and incorporate the reviewer's formatting suggestions.

*Hyperparameters.* We will indicate the choice of hyperparameters in our experiments. Specifically, for iForest, we used the default values suggested by the authors, namely sub-sampling size being 256 and number of trees being 100. For LODA, we used 100 projections with each projection using approximately $\sqrt{d}$ features.

*Theorem 3.5.* We thank the reviewer for checking the correctness and steps of our proofs, which we will expand to provide adequate details. In fact, Theorem 3.3 and 3.4 hold for most $p$ in $[0, 1)$, except for values that are too close (in a quantifiable way) to 0 or 1. The results in Theorem 3.5 remain valid, and we will expand the arguments in our proof to describe the effect of boundary cases (which only impacts the constants). The proof for Equation (3) of Theorem 3.5 will follow easily from Theorem 3.4 after the update. We will also set $k = \lfloor np \rfloor$, throughout.

*Comparison with refs. [1], [2] and [3].* We will comment on similarities and differences of our work with [1], [2] and [3], which are all essentially targeting the estimation of high-density points/regions. The reviewer has summarized them well. We would also add that we provide a new analysis of the minimal separation between the distribution of the normal and of the anomalous observation. (Finally, assumption A1 is eq. 24 in Theorem 12 of [14]).

*Software.* We apologize for the confusion. We will make our code available through a linked public github repository.

**Reviewer #2:**

*Table 1 & 2.* We will further emphasize the difference in interpretation between Table 1 and Table 2, and explain why we are using different evaluation metrics. Essentially, the failure rate serves as a metric for the average performance of different methods across the 20K synthetic datasets, whereas we believe practitioners may be more interested in obtaining raw AUC and AP scores on each real dataset.

*Practical Interpretation.* Figure 2 gives an example of a practical interpretation of Proposition 3.6: DTM will not make any mistake when the anomalies are sufficiently separated from the normal points, and identify the region for mis-classification when they are too close. We will include more illustration of the behavior of DTM in different scenarios, especially the ones where mistakes will be made near the boundary of the support of the distribution of the normal observations. We will also include additional commentary on the main theoretical results of Section 3.3.

*More precise description of the simulation results.* We will expand our comparisons of the different methods under various cases in our supplementary material. Each synthetic dataset is defined by a set of parameters (anomaly rate, difficulty, clusteredness, etc). We will indicate explicitly the performance of each method under each case, and comment on how the dataset parameters are related to the methods' performance.

**Reviewer #3:**

*How realistic are our assumptions and Proposition 3.6.* Our assumptions are fairly standard and generally regarded as mild in the literature on DTM and on geometric inference and are needed to rule out pathological cases. Yet, they capture a wide range of distributions. The assumptions in Proposition 3.6 encompass what are arguably some of the simplest instances of anomaly detection problems in fully non-parametric settings. We believe it is important to start with these cases to appreciate the difficulty of the task. We will add a remark.

*Theory for Ensemble Methods.* We are not aware of any theoretical analysis of ensemble methods, iForest and LODA in particular. Though this is a very important and broad question, it is also notoriously difficult and outside the scope of our paper.

*Sensitivity Analysis.* The only parameters for DTM are $k$ and $q$. We set $k$ to be the same as that for LOF for comparable results, and discussed the performance of DTM for $q = 1, 2, \infty$. We will include a discussion on sensitivity to the parameters and point the readers to the relevant literature.

*Related Literature.* We are thankful for the list of references, which we will definitely cite and comment on. We will mention the good performance of those sub-sampling and ensemble-based NN methods. It appears that the theoretical analyses from those references differ from ours significantly, in both assumptions and goals. We believe our theoretical results provide novel and complementary insights into the performance of KNN-based methods both for anomaly detection and for more general tasks.

**Reviewer #4:**

*Overlapping Support.* We agree: allowing for overlapping support is an interesting and more realistic case. However, to analyze it, it is necessary to add further assumptions on how the two distributions differ. In this paper, we consider a simpler case where we essentially leave the distribution of anomalous points unconstrained.

*Boxplots.* We agree: boxplots may be uninformative in this case. We used them as a coarse summary of the performance of 6 methods on 23 real datasets. Due to space limitations, we had to leave the full performance table (Table 3) in the supplementary file. We will add an extra row at the bottom of the table, indicating the average rank of each method, and conduct a pairwise Wilcoxon signed rank test.

*Complexity of the datasets in the experiments.* Please see response to Reviewer #2 point 3.

*Connection between empirical and theoretical studies.* We use section 2 as a motivation for the theoretical analysis in section 3, followed by a practical illustration of our theoretical results in section 3.4. We will be happy to hear and follow the reviewer suggestions on how to link these two parts seamlessly.

[Meta-Review · NeurIPS 2019]

Following the discussion, the referees converged on recommending acceptance.